# Differences in Airborne Particulate Matter Concentration in Urban Green Spaces with Different Spatial Structures in Xi'an, China

**Bo Jiang [1], Chang Sun [1], Sen Mu [1], Zixin Zhao [2], Yingyuan Chen [1], Yiwei Lin [1], Ling Qiu [1,*,†] and Tian Gao [1,*,†]**

[1] College of Landscape Architecture and Art, Northwest A&F University, No.3 Taicheng Road, Xianyang 712100, China; ASL6006@nwafu.edu.cn (B.J.); 2019059347@nwafu.edu.cn (C.S.); musen@nwafu.edu.cn (S.M.); cyydy@nwafu.edu.cn (Y.C.); linyiwei@nwafu.edu.cn (Y.L.)

[2] Accademia di Belle Arti di MILANO "Brera", Via Brera, 28, 20121 Milano, Italy; jennifer951129@163.com

* Correspondence: qiu.ling@nwsuaf.edu.cn (L.Q.); tian.gao@nwsuaf.edu.cn (T.G.); Tel.: +86-29-87080269 (L.Q.); +86-29-87082997 (T.G.)

† These two authors jointly supervised this work.

**Abstract:** With the acceleration of urbanization and industrialization, air pollution is becoming one of the most serious problems in cities. Urban green spaces, as "green infrastructure", are an important part of urban ecosystems for air purification. Therefore, 10 typical green spaces of urban parks in the city of Xi'an, China, were selected as study areas according to vegetation structure and species composition. Considering meteorological factors and time changes, the effects of the selected green spaces with different vegetation structures of different heights on the reduction in airborne particulate matter concentration were explored. The results showed that the following: (1) Temperature, relative humidity, wind speed, and air pressure had significant correlation with the concentration of airborne particulate matter at the different heights, and the correlations were the same at 1.5 m and 5 m. (2) After heating in winter, the concentration of airborne particulate matter with different particle sizes increased significantly. The concentration of airborne particulate matter showed different trends throughout the day, and the small particles ($PM_1$ and $PM_{2.5}$) had a trend of "lower in the morning and evening, and higher at noon", while the large particles ($PM_{10}$ and TSP) gradually decreased over time. (3) In the selected green spaces with different vegetation structure types, the concentration of airborne particulate matter below the canopy (1.5 m) was generally higher than that in the middle of the canopy (5 m), but the effects of reducing the concentration of airborne particulate matter were consistent at the different heights. (4) The adsorption capacity of $PM_1$ and $PM_{2.5}$ concentration was strong in the partially closed broad-leaved one-layered forest (PBO), and poor in the partially closed broad-leaved multi-layered forest (PBM). Partially closed broad-leaved multi-layered forest (PBM) and partially closed coniferous and broad-leaved mixed multi-layered forest (PMM) also had strong dust-retention effect on $PM_{10}$ and TSP, while closed broad-leaved one-layered forest (CBO) had a poor dust-retention effect. The results showed that the reduction effects of urban green spaces with different spatial structures on air particles were different, and were restricted by various environmental factors, which could provide a theoretical basis for the optimization of urban green space structure and the improvement of urban air quality.

**Keywords:** green space; airborne particulate matter; meteorological parameters; height

## 1. Introduction

With rapid economic development and urbanization, ecological problems are becoming increasingly prominent, and air pollution has become one of the most serious problems to be faced by the whole world—especially by largely urbanized and densely populated countries, such as China [1]. According to the state of China's Ecological Environment Bulletin 2019, 157 cities were not up to the desired standards of ambient air quality, accounting for 46.6% of the total number of cities in China, with the overall situation not being

optimistic. Particulate matter with small aerodynamic diameter is the primary pollutant, with an uneven surface and strong adsorption ability; it can make a variety of harmful substances in the air attach to its surface—such as polycyclic aromatic hydrocarbons (PAHs), bacteria, and viruses—and enter the human body through the respiratory tract, directly endangering human health, and resulting in an increase in death rates from lung cancer [2]. It has been confirmed that air pollution poses a serious threat to human quality of life, and decreases the life expectancy of inhabitants who live in urban areas [3].

Depending on its aerodynamic diameter, airborne particulate matter can be classified into total suspended particulate matter (TSP; diameter $\leq 100$ μm), inhalable particulate matter (PM$_{10}$; diameter $\leq 10$ μm), fine particles (PM$_{2.5}$; diameter $\leq 2.5$ μm), and ultrafine particles (PM$_1$; diameter $\leq 1$ μm). PM originates from both natural sources—such as wildfires and dust storms [4]—and anthropogenic activities, such as mining and the burning of fossil fuels [5,6] The burning of fossil fuels for heating in winter releases waste, which provides direct or indirect conditions for the growth of airborne particulate matter—especially in the northern part of China. Many practical strategies have been proposed by the public to reduce particulate pollution in cities. Environmental laws have been passed to lower the amounts of toxic emissions from factories, modify energy resource structures, and limit vehicle numbers [7].

However, the control of air pollution is not an overnight action, but a long and arduous process. Urban green spaces, as "green infrastructure", are an important part of urban ecosystems, which play a significant role in improving the ecological environment, beautifying and optimizing the living environment. Several studies have shown that urban areas with high green coverage helped to reduce the concentration of airborne particulate matter, and often had a negative correlation with the latter [8,9]. It was found that the size, shape, and microstructure of plant leaves in green spaces—such as leaf surface roughness, waxy layer, and leaf hair—had significant effects on the capture of airborne particulate matter [10]. Therefore, using green spaces to purify dust in the air is a practical and effective measure, and the dust-retention ability of plants has become an important index of plant selection in urban green spaces [11].

The dust-catching ability of different plants in urban green spaces varies greatly. Some studies have found that the dust-catching ability of different plant communities is as follows: arbors > shrubs > herbs [12,13]. Tall trees mainly retard and filter the particulate matter and drifting particulate matter from the outside world, while shrubs and grasses can effectively intercept the particulate matter from the ground [14]. Przybysz et al. found that urban meadows accumulate PM from the ambient air more effectively than tradition lawns [15]. Terzaghi et al. found that an arbor–shrub–grass planting model can better reduce the concentration of airborne particulate matter [16]. However, McDonald et al. showed that more particulate matter will be retained in the compound structure of green spaces with a combination of trees, shrubs, and grass [17]. Therefore, these inconsistencies require further attention in order to expose the underlying factors. According to the study of Gao et al., vegetation structure plays a significant role in regulating air pollution, and the leaf dust retention of even the same species of plants was different in the urban green spaces with different vegetation structures (e.g., enclosed green spaces had lower dust retention than open green spaces) [18]. Moreover, Selmi et al. claimed that the dust retention in the "low" position of the same plant leaves was significantly higher than that in the "high" and "middle" positions [19]; in other words, different vertical heights of plants in the green spaces also showed great differences in their ability to reduce the concentration of airborne particulate matter. Therefore, it is necessary to systematical study the ability of green spaces with different spatial structures—including vegetation heights—in order to reduce the concentration of airborne particulate matter.

In addition, meteorological environmental parameters in urban green spaces also affect the concentration of particulate matter in different sections of vegetation. Wind speed, temperature, relative humidity, and air pressure affect the diffusion and settlement of particles with different sizes [20,21]. Wind speed plays an important role in horizontal

transmission and dilution diffusion [22]. Changes in temperature affect convection in the vertical direction of the atmosphere which, in turn, affects the concentration of airborne particulate matter [23]. Changes in relative humidity also increase the concentration of fine particles [24]. When the air pressure change is obvious, the atmosphere is in an unstable state, facilitating the vertical diffusion of pollutants [25]. Given the above, the influence of meteorological factors dust retention should also be comprehensively taken into account.

Therefore, combined with the meteorological parameters, the concentration changes of particles with different sizes in urban green spaces with different vegetation structures and types at different heights should be explored in order to provide the optimal urban green space planning for future air quality improvement. The main objectives of this study are to investigate the following:

- The factors of meteorological parameters, monitoring time, vegetation structure, and vegetation height influencing the concentration of airborne particulate matter;
- The differences in the concentration distribution of airborne particulate matter in urban green spaces with different vegetation structures at different heights.

## 2. Materials and Methods

### 2.1. Study Area

Xi'an, the capital of Shaanxi Province, is located in the central part of the Guanzhong Plain, between 107.40°–109.49° E and 33.42°–34.45° N, bordering the Wei River in the north and the Qinling Mountains in the south of China; it is one of the important birthplaces of the Chinese civilization and nation, and the starting point of the Silk Road. Xi'an belongs to the semi-humid continental monsoon climate of the warm temperate zone, with distinct four seasons: spring is variable, summer is hot and rainy, autumn is cool and rainy, and winter is dry and cold with little rain or snow [26].

### 2.2. Classification and Selection of Urban Green Spaces

Through Google satellite image interpretation and field investigation, combined with the characteristics of urban green spaces in the city of Xi'an (Figure 1), the green spaces were first divided according to their spatial vegetation structure and species composition. By using the LAI-2200 Plant Canopy Analyzer and a fisheye camera (Figure 2), the canopy cover ratios of trees and shrubs were first divided into open green spaces (<10% canopy cover of trees/shrubs), partially open green spaces (10–40% canopy cover of trees/shrubs), partially closed green spaces (40–70% canopy cover of trees/shrubs), and closed green spaces (>70% canopy cover of trees/shrubs). According to the species composition, open green spaces and partially open green spaces were then divided into two subtypes—lawn, and grassland—while partially closed green spaces and closed green spaces were divided into three subtypes: coniferous forest, broad-leaved forest, and coniferous and broad-leaved mixed forest. Partially closed green spaces and closed green spaces were further subdivided into two types—one-layered forest, and multi-layered forest—forming a set of unified standard urban green space classification systems (Table 1) [27].

According to the actual situation of the city of Xi'an, 10 typical and abundant sample types of sites were selected, including open lawn (O), partially open green space (PO), partially closed broad-leaved one-layered forest (PBO), partially closed broad-leaved multi-layered forest (PBM), partially closed coniferous and broad-leaved mixed multi-layered forest (PMM), closed broad-leaved one-layered forest (CBO), closed broad-leaved multi-layered forest (CBM), closed coniferous one-layered forest (CCO), closed coniferous and broad-leaved mixed one-layered forest (CMO), and closed coniferous and broad-leaved mixed multi-layered forest (CMM). Two or three sample plots of each type were selected as duplicates, and two public squares with hard pavement were taken as the control groups (CK), which were quite open and covered by green space around the square. A total of 26 sample plots were finally selected (Figure 3).

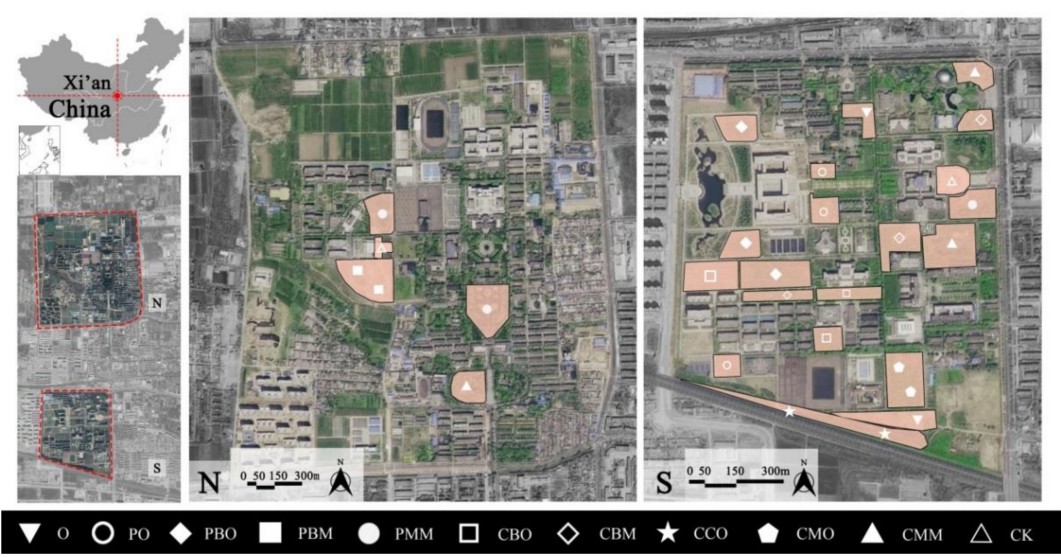

**Figure 1.** Map showing sampling locations and their repetition.

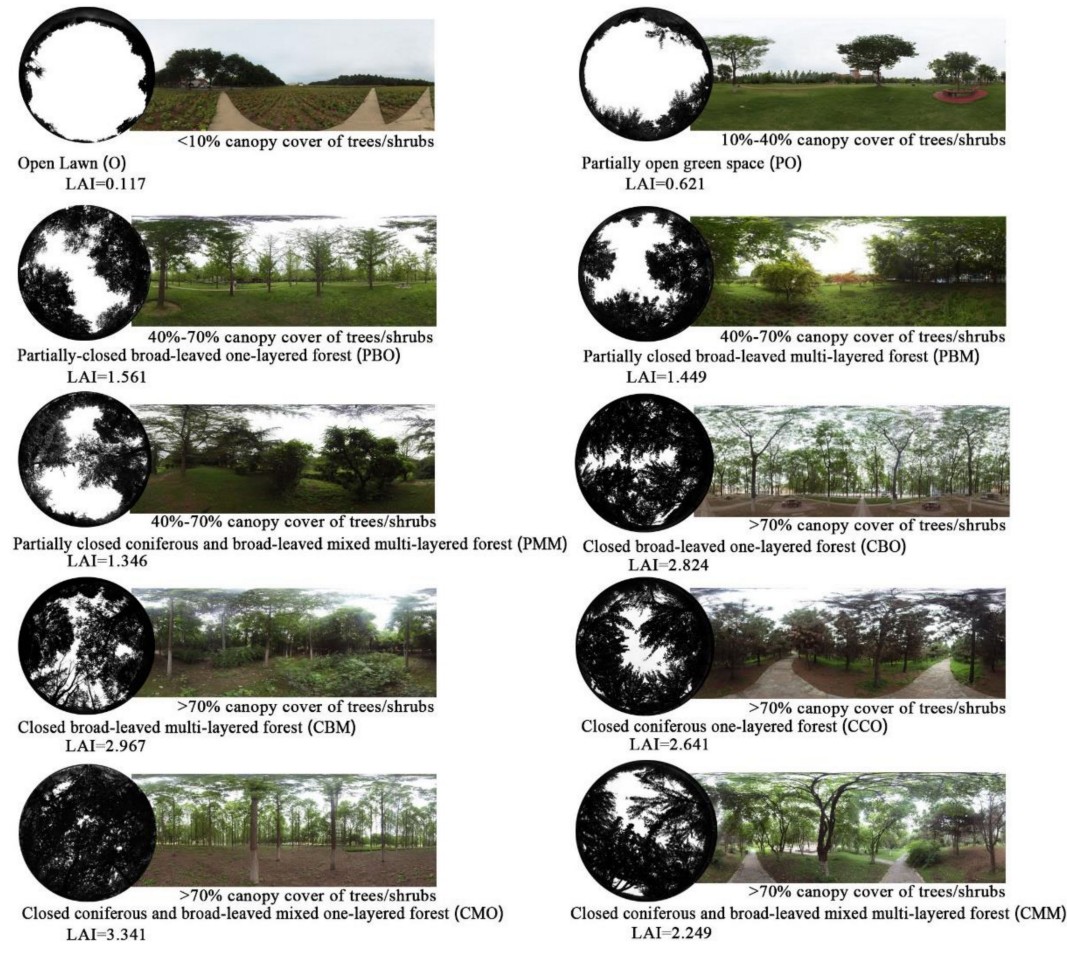

**Figure 2.** Fisheye camera view of urban green space classification.

**Table 1.** Classification with three levels shown for urban green spaces.

| | Horizontal Structure | Species Composition | Vertical Structure |
|---|---|---|---|
| Urban green spaces | Open green spaces (<10% canopy cover of trees/shrubs) | Lawn mainly dominated by *Cynodon dactylon* Grass flowers mainly dominated by *Veronica persica* | – |
| | Partially open green spaces (10–40% canopy cover of trees/shrubs) | Lawn mainly dominated by *Arachis hypogaea* Grass flowers mainly dominated by *Oxalis corymbosa* | |
| | Partially closed green spaces (40–70% canopy cover of trees/shrubs) | Broad-leaved trees mainly dominated by *Melia azedarach* Coniferous trees mainly dominated by *Pinus tabuliformis* Mixed plants mainly dominated by *Ligustrum sinense* and *Cedrus deodara* | One-layered Multi-layered |
| | Closed green spaces (>70% canopy cover of trees/shrubs) | Broad-leaved trees mainly dominated by *Platanus orientalis* Coniferous trees mainly dominated by *Picea asperata* Mixed plants mainly dominated by *Koelreuteria paniculata* and *Platycladus orientalis* | One-layered Multi-layered |

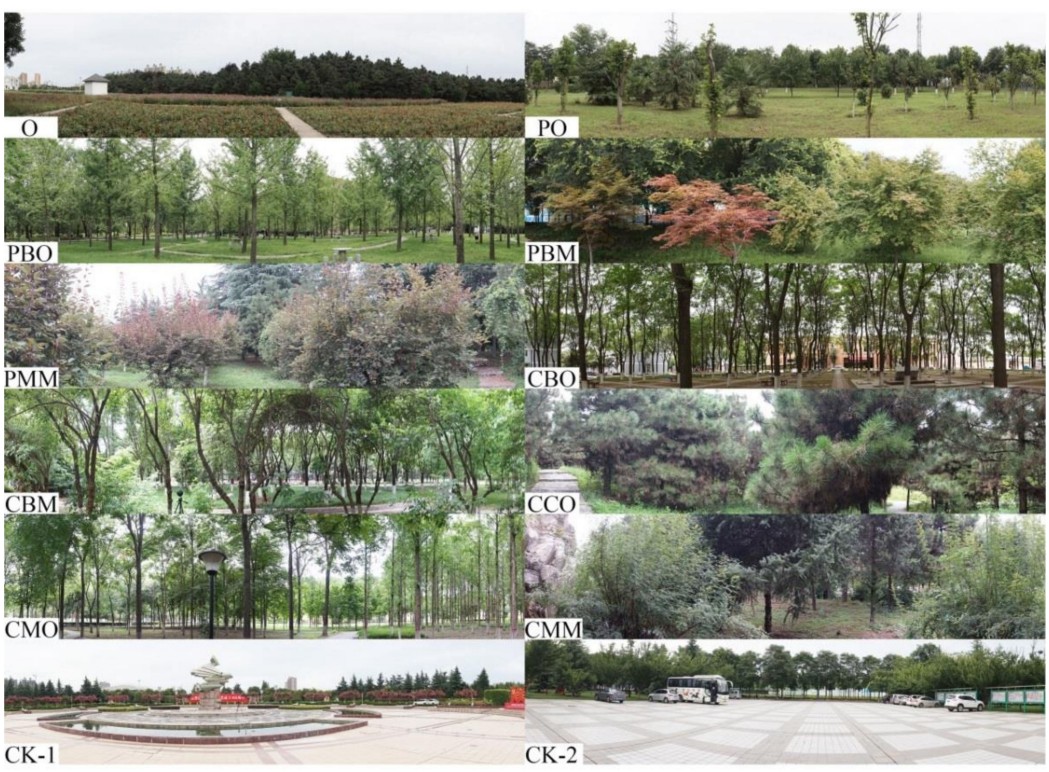

**Figure 3.** Panoramic views of 10 different vegetation structures (O: open lawn; PO: partially open green space; PBO: partially closed broad-leaved one-layered forest; PBM: partially closed broad-leaved multi-layered forest; PMM: partially closed coniferous and broad-leaved mixed multi-layered forest; CBO: closed broad-leaved one-layered forest; CBM: closed broad-leaved multi-layered forest; CCO: closed coniferous one-layered forest; CMO: closed coniferous and broad-leaved mixed one-layered forest; CMM: closed coniferous and broad-leaved mixed multi-layered forest; CK: the control groups).

### 2.3. Field Monitoring

In order to combine the influence of meteorological parameters on reducing the concentration of airborne particulate matter in urban green spaces, meteorological parameters (i.e., temperature, relative humidity, wind speed, and air pressure) were monitored in sunny, windy, or calm weather. The meteorological parameters were monitored through the use of handheld weather stations (Kestrel 5500), and the concentrations of airborne particulate matter (TSP, $PM_{10}$, $PM_{2.5}$, and $PM_1$) were measured with handheld particle counters (Aerocet 831) in the selected plots. In the grid pattern of each sample plot, four locations were uniformly selected as sampling points to represent the typical sample types. The monitors were set up at heights of 1.5 m and 5 m in each sample plot—1.5 m is the average height at which human respiration occurs, while 5 m is the average height at the middle of the plant canopy; thus, it is convenient to compare the effects of different heights of vegetation on reducing the concentration of airborne particulate matter. In addition, the surrounding environment of each sampling plot was consistent, without pollution sources, and at least 20 m away from the road and the water edge as a buffer, so as to avoid other factors affecting the experimental results. The data were collected during the three months from 1 October 2020 to 31 December 2020, which belonged to the pre-heating and heating periods. Two days were selected from the beginning and the end of each month, and three time periods per day were taken—from 8:00 to 10:00, 12:00 to 14:00, and 16:00 to 18:00. All sampling sites were monitored sequentially at the same time.

### 2.4. Statistical Analysis

In this study, Microsoft Office Excel 2010 software was used for all data recording and collection. Generalized linear analysis and correlation analysis were used in the statistical software package IBM SPSS Statistics 19 to model all variables, in order to explore whether each variable had a significant effect on airborne particulate matter, and to identify how each variable affected airborne particulate matter concentration. The acceptable significance level was $p < 0.05$.

## 3. Results

### 3.1. Effects of the Dominated Factors on Airborne Particulate Matter

In this study, it was found that the meteorological parameters, monitoring time, vegetation structure, and height had significant effects on the concentrations of $PM_1$ and $PM_{2.5}$ (Table 2). Some meteorological parameters had a certain influence on the concentrations of $PM_{10}$ and TSP. Except for temperature, vegetation structure, and air pressure, the other factors had significant effects on $PM_{10}$ and TSP concentrations.

**Table 2.** Variance analysis of factors affecting PM concentration.

| Parameters | Df | $PM_1$ | | $PM_{2.5}$ | | $PM_{10}$ | | TSP | |
|---|---|---|---|---|---|---|---|---|---|
| | | F-Value | *p*-Value | F-Value | *p*-Value | F-Value | *p*-Value | F-Value | *p*-Value |
| Temperature | 1 | 2537.40 | 0.000 | 1979.89 | 0.000 | 2.88 | 0.090 | 38.45 | 0.000 |
| Relative humidity | 1 | 843.40 | 0.000 | 865.56 | 0.000 | 4.97 | 0.026 | 50.14 | 0.000 |
| Wind velocity | 1 | 247.26 | 0.000 | 224.79 | 0.000 | 9.12 | 0.003 | 5.73 | 0.017 |
| Air pressure | 1 | 7.42 | 0.006 | 4.79 | 0.029 | 1.43 | 0.023 | 2.16 | 0.142 |
| Pre-heating and heating periods | 1 | 3086.40 | 0.000 | 3245.85 | 0.000 | 8.56 | 0.003 | 4.81 | 0.028 |
| Monitoring time | 1 | 298.35 | 0.000 | 446.50 | 0.000 | 3.73 | 0.035 | 0.49 | 0.048 |
| Vegetation structure | 10 | 27.35 | 0.000 | 10.85 | 0.001 | 3.25 | 0.071 | 2.00 | 0.015 |
| Height | 1 | 74.24 | 0.000 | 10.97 | 0.001 | 23.96 | 0.000 | 29.34 | 0.000 |

Df: degree of freedom; F-Value: variance test volume; *p*-Value: significant test of regression equation.

### 3.2. Effects of Meteorological Parameters on Airborne Particulate Matter

The results showed that the meteorological parameters—including temperature, relative humidity, wind speed, and air pressure—had significant correlations with the concentration of airborne particulate matter at different vertical heights, and the correlations of meteorological parameters with the concentration of airborne particulate matter were consistent at the heights of 1.5 m and 5 m (Table 3).

**Table 3.** Correlation analysis of meteorological parameters and airborne particulate matter concentrations.

| Height | Project | Spearman Correlation Test | | | |
|---|---|---|---|---|---|
| | | Temperature (°C) | Humidity (%) | Wind Speed (m/s) | Air Pressure (mpa) |
| 1.5 m | $PM_1$ ($\mu g/m^3$) | −0.087 ** | 0.175 ** | −0.224 ** | 0.212 ** |
| | $PM_{2.5}$ ($\mu g/m^3$) | −0.253 ** | 0.251 ** | −0.188 ** | 0.311 ** |
| | $PM_{10}$ ($\mu g/m^3$) | −0.147 ** | 0.026 | −0.051 * | −0.021 |
| | TSP ($\mu g/m^3$) | −0.110 ** | 0.040 | −0.040 | −0.087 ** |
| 5 m | $PM_1$ ($\mu g/m^3$) | −0.177 ** | 0.168 ** | −0.229 ** | 0.028 |
| | $PM_{2.5}$ ($\mu g/m^3$) | −0.169 ** | 0.250 ** | −0.193 ** | 0.098 ** |
| | $PM_{10}$ ($\mu g/m^3$) | −0.143 ** | 0.034 | −0.024 | −0.056 * |
| | TSP ($\mu g/m^3$) | −0.131 ** | 0.020 | −0.038 | −0.110 ** |

**: At level 0.01 (two-tailed), the correlation was significant; *: at level 0.05 (two-tailed), the correlation was significant.

During the October–December period of measurement, the temperature varied from −3.4 to 26.4 °C. The effect of temperature on the concentration of airborne particulate matter at different heights was consistent, showing a significant negative correlation—that is, the higher the temperature, the lower the concentration of airborne particulate matter (Figure 4a,b). Humidity had a significant positive correlation with the concentration of airborne particulate matter (Figure 4c,d). With the increase in relative humidity, $PM_1$ and $PM_{2.5}$ were more likely to condense, leading to an increase in the concentration of airborne particulate matter. In this study, the variation range of wind speed was 0–3.5 m/s. With the increase in wind speed, the diffusion of airborne particulate matter concentration was promoted, thus reducing the airborne particulate matter concentration (Figure 4e,f). The vertical atmospheric pressure ranged from 955.2 to 996.4 mPa. The change in air pressure caused different changes in the concentration of airborne particulate matter with different particle sizes (Figure 4g,h); it was positively correlated with the concentrations of fine particulate matter $PM_1$ and $PM_{2.5}$, but negatively correlated with the concentrations of particulate matter $PM_{10}$ and TSP—that is, the higher the air pressure, the lower the concentrations of $PM_{10}$ and TSP.

### 3.3. Effects of Diurnal Variation before and after the Heating Period on Airborne Particulate Matter

The generalized linear models of time—before and after heating—and the concentration of airborne particulate matter were established for statistical analysis, and meteorological parameters were taken into account. The results showed that time in both the pre-heating and heating periods had significant effects on the concentration of airborne particulate matter ($p < 0.01$). The concentration of airborne particulate matter increased significantly after heating, and the highest concentration reached 10 times that of the pre-heating period (Figure 5). The concentrations of airborne particles with different particle sizes were significantly different in different time periods. The smaller particles, such as $PM_1$ and $PM_{2.5}$, showed a trend of "lower in the morning and evening, and higher at noon"; however, $PM_{10}$ and TSP showed a trend of gradual decrease (Figure 6).

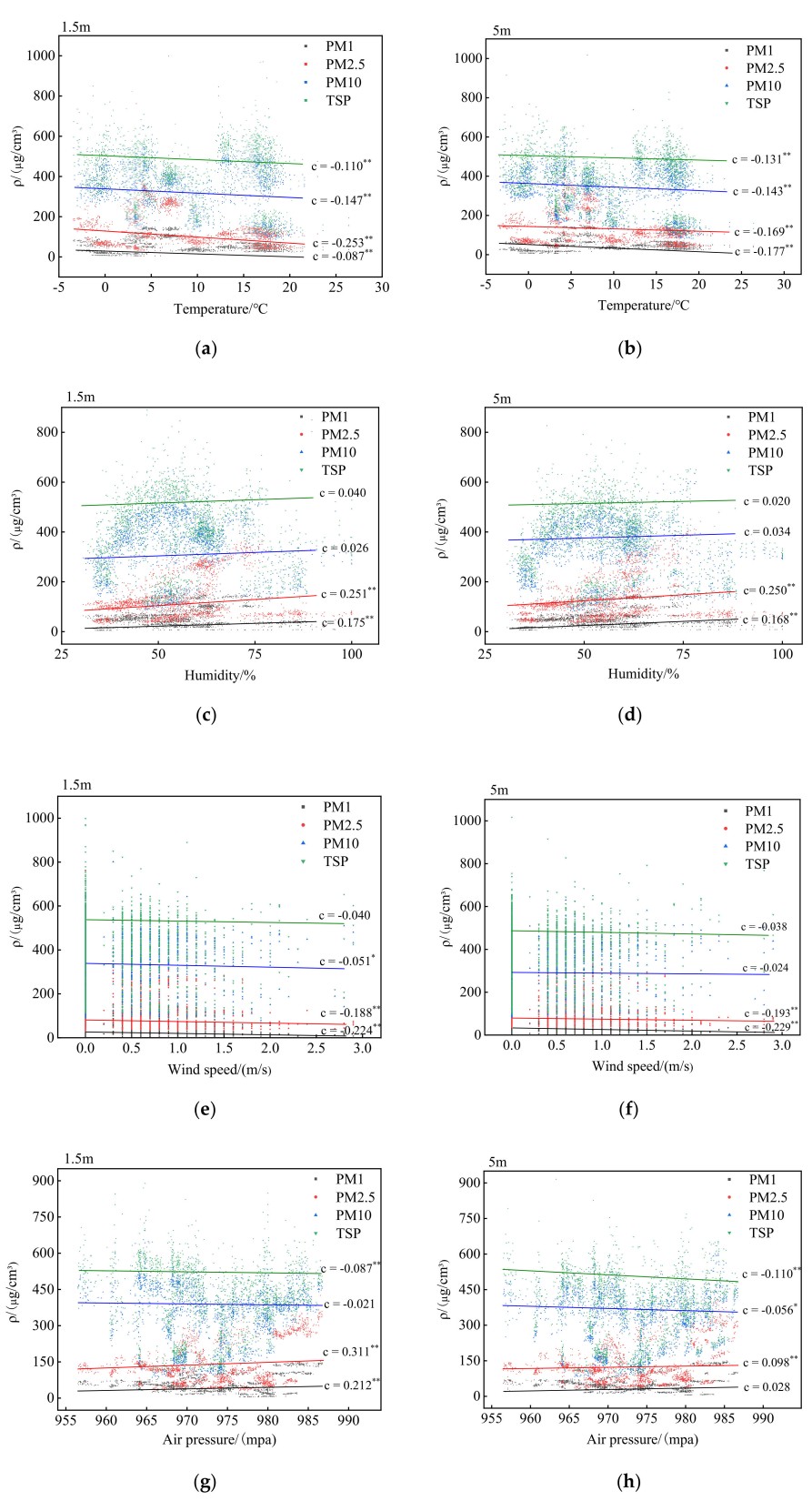

**Figure 4.** The relationship between meteorological factors and PM concentration in the different heights: (**a**,**b**) effect of temperature on the concentration of airborne particulate matter; (**c**,**d**) effect of humidity on the concentration of airborne particulate matter; (**e**,**f**) effect of wind speed on the concentration of airborne particulate matter; (**g**,**h**) effect of air pressure on the concentration of airborne particulate matter; "c" stands for correlation coefficient; **: at level 0.01 (two-tailed), the correlation was significant; *: at level 0.05 (two-tailed), the correlation was significant.

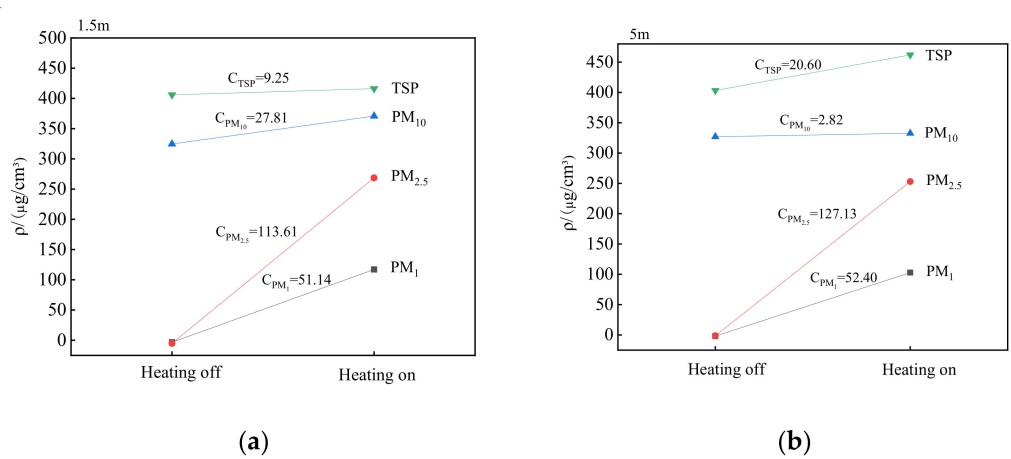

**Figure 5.** Changes in airborne particulate matter concentration before and after heating at different heights. Note: C represents the generalized linear analysis coefficient of the change in airborne particulate matter concentration before and after heating. The absolute value of C represents the difference between the factor level and the population mean. (**a**) Changes in airborne particulate matter concentration before and after heating at 1.5m height; (**b**) Changes in airborne particulate matter concentration before and after heating at 5m height.

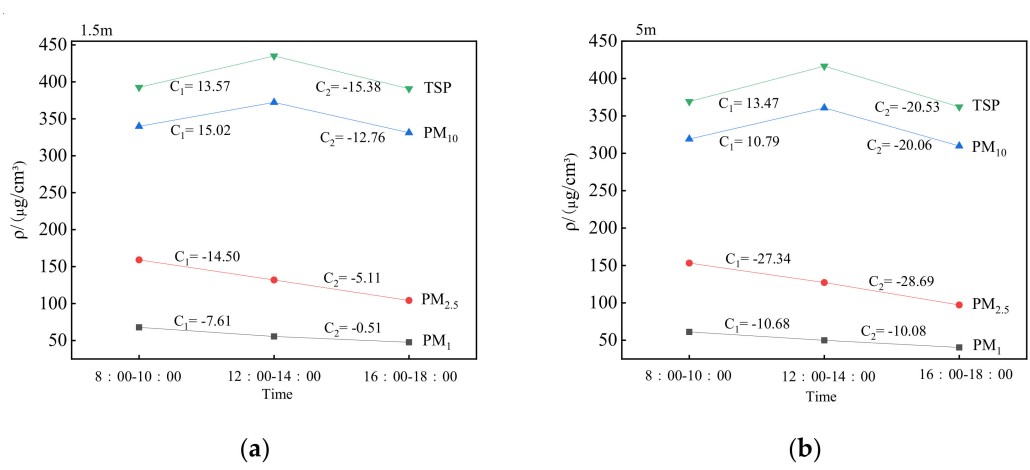

**Figure 6.** Influence of different time periods on airborne particulate matter at different heights. Note: C1 represents the generalized linear analysis coefficient of the change in airborne particulate matter concentration in the morning and noon. C2 represents the generalized linear analysis coefficient of the change in airborne particulate matter concentration in the noon and evening. The absolute value of C represents the difference between the factor level and the population mean. (**a**) Influence of different time periods on airborne particulate matter at 1.5m height; (**b**) Influence of different time periods on airborne particulate matter at 5m height.

### 3.4. Effects of Different Vegetation Structures on the Concentration of Airborne Particulate Matter

The results showed that there were significant differences in height and airborne particulate matter concentration among the different vegetation structures (Figure 7), and the concentration of airborne particulate matter below the canopy (1.5 m) was much higher than that in the middle of the canopy (5 m). There was no significant difference in the concentration of airborne particulate matter between the control groups (CK) and the different vegetation structure types.

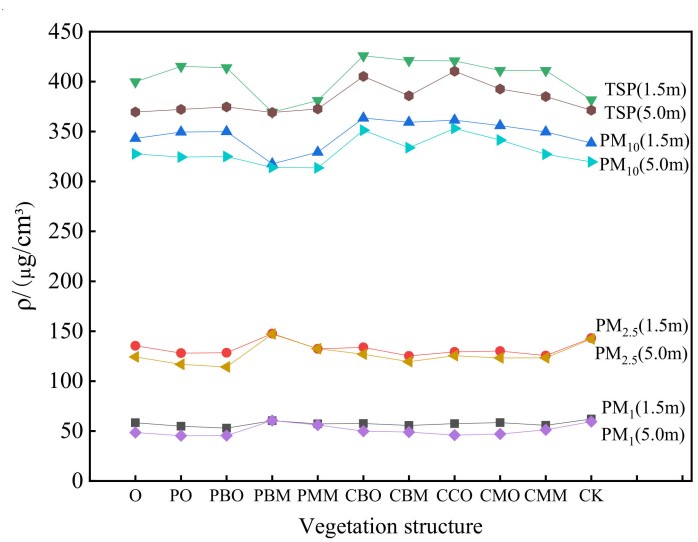

**Figure 7.** Distribution of mean changes in airborne particulate matter by vertical height.

In the sample plots with different vegetation structures, the concentrations of airborne particulate matter with different particle sizes at different heights were different (Figure 7). However, the concentration was almost the same in the partially closed broad-leaved multi-layered forest (PBM). The concentration of $PM_1$ in the closed coniferous one-layered forest (CCO) had the largest difference, reaching 11.60 $\mu g/cm^3$. In other vegetation structure plots, the differences in $PM_1$ concentration at different heights were small. The difference in $PM_{2.5}$ concentration was mainly concentrated in the open (O) and partially open (PO) green spaces, but in the partially closed broad-leaved one-layered forest (PBO), the concentration at 1.5 m height was much higher than that at 5 m height, and the difference value could reach 14.27 $\mu g/cm^3$. In addition to partially closed broad-leaved multi-layered forest (PBM), the largest difference in $PM_{10}$ concentration was found in other multi-layered structural forests, such as PMM, CBM, and CMM. Except for the PBM, partially closed coniferous and broad-leaved mixed multi-layered forest (PMM) and closed coniferous one-layered forest (CCO), the difference in TSP concentration in other plots was more than 10 $\mu g/cm^3$.

Moreover, there were differences in the reduction effect of green spaces with different spatial vegetation structures on the concentration of airborne particulate matter with different particle sizes. At the height of 1.5 m, the reduction effects of the 10 different vegetation structures on $PM_1$ and $PM_{2.5}$ concentrations were similar, but the reduction effects on $PM_{10}$ and TSP concentrations were significantly different; among them, the partially closed broad-leaved one-layered forest (PBO) and the partially open green spaces (PO) had the most significant negative effects on $PM_1$ and $PM_{2.5}$, as well as the strongest adsorption capacity. The closed broad-leaved one-layered forest (CBO) had a positive effect on $PM_{10}$ and TSP, and the concentrations of $PM_{10}$ and TSP were the highest in this type of green space. In the partially closed broad-leaved multi-layered forest (PBM), the concentrations of $PM_{10}$ and TSP were the lowest, and the dust-retention ability was strong (Figure 8a).

At the height of 5 m, there were significant differences in $PM_1$ reduction among the 10 green spaces with different vegetation structures (Figure 8b). Except for the partially closed broad-leaved multi-layered forest (PBM) and the partially closed coniferous and broad-leaved mixed multi-layered forest (PMM), $PM_1$ concentrations were negatively affected by other vegetation structures. In particular, the partially closed broad-leaved one-layered forest (PBO) had the best dust-retention effect for $PM_1$ and $PM_{2.5}$. In addition, the partially closed broad-leaved multi-layered forest (PBM), partially closed coniferous and broad-leaved mixed multi-layered forest (PMM), and closed broad-leaved one-layered forest (CBO) had positive impacts on $PM_{2.5}$ concentration—that is, $PM_{2.5}$ concentration was higher in these types of green spaces. In the closed broad-leaved one-layered forest

(CBO), closed coniferous one-layered forest (CCO,) and closed coniferous and broad-leaved mixed one-layered forest (CMO) there were significant differences in $PM_{10}$ concentration reduction, while the other seven types of green spaces had moderate and similar reduction effects on $PM_{10}$ concentration. In the closed coniferous one-layered forest (CCO) and the closed broad-leaved one-layered forest (CBO), TSP concentration reached the maximum value, and dust-retention ability was weak; however, the partially closed broad-leaved multi-layered forest (PBM) had a strong reduction effect on $PM_{10}$ and TSP.

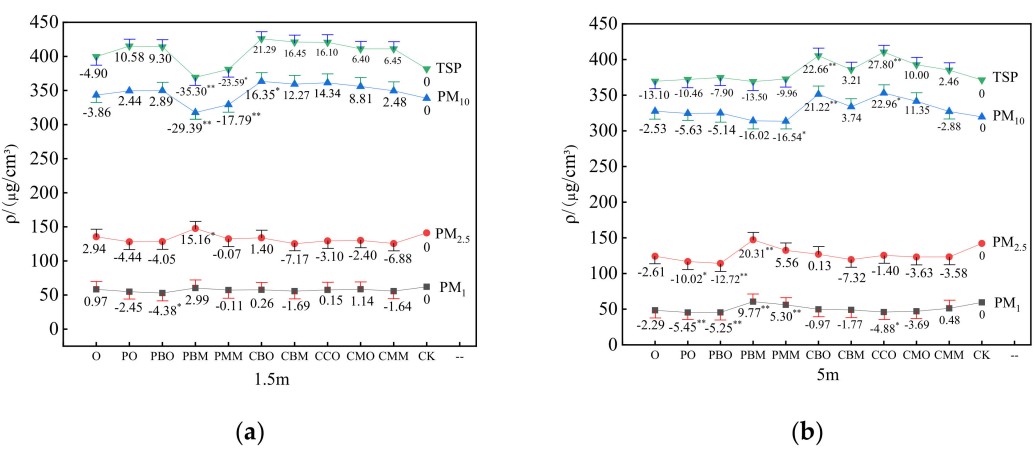

**Figure 8.** Comparison of airborne particulate matter concentrations in vegetation structures at different vertical heights. Note: The values in the figure represent the correlation coefficients of generalized linear analysis between green spaces with different vegetation structures and concentrations of airborne particulate matter with different particle sizes—that is, the distance between different factor levels and the population mean. **: At level 0.01 (two-tailed), the correlation was significant; *: At level 0.05 (two-tailed), the correlation was significant. (**a**) Comparison of airborne particulate matter concentrations in the different vegetation structures at 1.5m height; (**b**) Comparison of airborne particulate matter concentrations in the different vegetation structures at 5m height.

In general, the reduction effects of different vegetation structures on the concentrations of particulate matter with different particle sizes were significantly different, but the reduction effect was consistent at different heights. Regardless of whether the height was 1.5 m or 5 m, the partially closed broad-leaved one-layered forest (PBO) had the strongest dust-retention ability for small $PM_1$ and $PM_{2.5}$ particles, followed by partially open green spaces (PO). The concentrations of $PM_1$ and $PM_{2.5}$ were the highest in the partially closed broad-leaved multi-layered forest (PBM). For the large $PM_{10}$ and TSP particles, the partially closed broad-leaved multi-layered forest (PBM) and the partially closed coniferous and broad-leaved mixed multi-layered forest (PMM) had strong adsorption capacity and a good dust-retention effect, while the closed broad-leaved one-layered forest (CBO) structure had a weak reduction effect.

## 4. Discussion

The results show that the concentrations of airborne particulate matter with different particle sizes in urban green spaces were all affected by meteorological parameters, monitoring time, vegetation structure, and height, indicating that the control of air pollution by urban green spaces is indeed a complex process, due to the synthetic action of these factors. Therefore, more factors and their synergies should be taken into consideration for improving air quality via urban green spaces in the future.

### 4.1. The Influence of Meteorological Parameters on the Concentration of Airborne Particulate Matter

Our results showed that temperature, relative humidity, wind speed, and air pressure had significant differences in their effects on the concentration of airborne particulate matter.

With the increase in temperature, air convection in the vertical direction was more frequent. Such gas circulation exchange accelerated the transport of airborne particulate matter, which was conducive to reducing the concentration of airborne particulate matter [28]. Airborne particulate matter concentration and relative air humidity showed good consistency. When the relative humidity of the air increased, the concentration of airborne particulate matter also increased. The increase in humidity made the moisture in the air increase, the particles become moist, and the weight of the particles increase, thus reducing the diffusion of the particles, so that the particles gathered to a certain extent, leading to an increase in the concentration of the particles in the air [29]. When the relative humidity increased to a certain extent, the wet sedimentation increased, and then the concentration of airborne particulate matter decreased. In addition, the increase in wettability and relative humidity can trigger certain biological particle emission mechanisms, such as active wet ejection of fungal spores or hygroscopic expansion of pollen fragmentation, thereby increasing the concentration of airborne particulate matter around the plants [30]. The influence of wind speed on airborne particulate matter varied with the surrounding environment. In this study, the conditions of sunny, breezes, or no wind were selected for monitoring (the mean wind speed was 0.31 m/s), and it was still found that wind speed had a significant negative correlation with the concentration of airborne particulate matter. The greater the wind speed, the more conducive to the spread of pollutants, and the less the $PM_{2.5}$ concentration. Small wind or no wind, an obvious inversion layer limited vertical movement of the low-altitude atmosphere [31]. In airborne particulate matter stuck at low heights or close to the ground, fog/haze weather appeared frequently. The airborne particulate matter was difficult to spread and not conducive to dilution to the periphery, which facilitated the formation of local accumulated airborne particulate matter, subjecting the air quality to a high concentration of pollution [32]. The reason for this was that low-speed wind cannot carry away water vapor; thus, the increase in humidity was conducive to the formation of haze [33]. When the air pressure was lower, the concentrations of $PM_1$ and $PM_{2.5}$ decreased accordingly. After the change in air pressure, the particles converged to the middle in the horizontal direction, and moved upward in the vertical direction. At the same time, the wind speed in the horizontal direction was low, and the horizontal diffusion conditions of particles were unfavorable [34].

### 4.2. The Influence of Time on the Concentration of Airborne Particulate Matter

It can be found that time has a significant effect on the concentration of airborne particulate matter. Airborne particulate matter concentration in the heating period was much higher than that in the pre-heating period. The causes of this phenomenon were diverse. In autumn and winter, waste from burning fossil fuels in northern China contributed directly to the increase in the concentration of particulate matter in the air [35]. At the same time, the decrease in temperature and precipitation was not conducive to the settling of airborne particulate matter [36]. In the winter heating period, broad-leaved trees shed a large number of leaves, and only some evergreen broad-leaved trees and conifers act as dust traps; thus, this reduced the adsorption capacity of particles, resulting in a higher concentration of airborne particulate matter. PM is a kind of aerosol substance, and this kind of airborne particulate matter has a certain gravity effect. The daily variation of airborne particulate matter with small particle size reached the maximum value between 12:00 and 14:00 at noon, which may be related to human activities, and the flow of people was large at noon [37]. The convective exchange and vertical diffusion of the atmosphere were strengthened, and the thickness of the mixing layer increased [38]. At night, after sedimentation, the larger diameter particles accumulated continuously near the ground. At the same time, solar radiation could also be a factor in the diurnal variation of airborne particulate matter concentration on sunny days [39]. With the emergence of solar radiation during the day, the ground temperature rose and formed a warm air mass near the ground. The warm air mass near the ground rose with the particles, so the concentration of particles

near the ground began to increase at noon, and reached its minimum in the evening with the influence of temperature and light.

*4.3. The Influence of Vegetation Structure on the Concentration of Airborne Particulate Matter at Different Heights*

Among the 10 types of green spaces in this study, the distributions of airborne particulate matter in different vegetation structures were different. Our results showed that the concentration of airborne particulate matter in the lower part of the canopy was much higher than that in the middle part of the canopy, which is consistent with the findings of Chan [40]. Due to the different underlying surface of the sample sites, with the increase in height, the atmospheric humidity first decreased, and the concentration of airborne particulate matter also decreased [41]. In autumn and winter, when the temperature dropped, fine particles had a strong suspension capacity in the atmosphere, and were also affected by various factors at different heights, such as leaf adsorption and branch blocking [42]. Coarse particulate matter was larger in size, and had obvious sedimentation in the atmosphere. The concentration near the ground was higher than that at the height of the plant canopy. With the passage of time, after the heating began, the branches of trees would also block the diffusion of particles. At the same time, when the temperature dropped, the particles kept colliding and undergoing friction, condensation, settlement, etc., and their concentration gradually increased [43].

Green spaces with different special structures had different effects on the reduction in the concentration of airborne particulate matter with various particle sizes. The larger the particle size, the stronger the reduction ability of the green spaces. The reason for this may be that plants could use their special micromorphological structures to play a certain retention role [44]. The quantity ratio and volume ratio of $PM_{10}$ retained on leaf surfaces are often much higher than those of $PM_{2.5}$ [45], indicating that plants have a stronger effect on reducing airborne particulate matter with larger particles. The density of particulate matter was higher than that of the single-layer structure in the green spaces, which may be related to the higher plant density in the green spaces, along with the poor ventilation conditions in the forest, which was not conducive to the transport and diffusion of airborne particulate matter [46]. Sehmel et al. found that when the dust-containing air flow passed through the tree crown, some dust with larger particles was blocked by the branches/leaves and fell, and could be returned to the air due to the action of wind and other external forces, while the other part remained on the surface of the branches and leaves [47]. The amount of PM gathered on leaves depends on the quantity, size, and morphology of the leaves, and can also be increased by the presence of epicuticular waxes, in which PM can become stuck or immersed [48]. In the closed multi-layered structural forest, the vertical structure of the vegetation was complex and there were many plant species, which to some extent hinders the airborne particulate matter from settling to the ground [49]. The particles adsorbed on the surfaces of plant leaves were only temporarily retained, and were prone to bouncing back and then being suspended in the atmosphere, thus increasing the concentration of airborne particulate matter [50]. Urban flowering meadows are more structurally and botanically diverse than lawns. Influenced by natural ecosystems, urban flowering meadows are mowed less frequently, thus reducing the emission of particulate matter into the air [15]. However, in the open green spaces dominated by lawns, the settling effect of airborne particulate matter in the air was less hindered, and the airborne particulate matter directly settled on the ground due to the effect of gravity [51]. In addition, mosses showed a higher capability of trapping atmospheric particulate matter than certain trees [52]. Therefore, different vegetation structures had different distributions of airborne particulate matter at different heights of vegetation.

*4.4. Limitations and Future Study*

The shortcomings of this study are that there was a set period of monitoring the concentration of particulate matter over the spatial structure, and the setting of the height gradient was limited. The next step should be to increase the monitoring time and vegetation height for further study, in order to provide an optimal urban green space planning scheme for future reduction in the concentration of airborne particulate matter.

**5. Conclusions**

Taking the urban green spaces with different spatial structures in Xi'an as the study area, and considering meteorological parameters, this study quantitatively compared the effects of monitoring time, spatial structure, and vegetation height on the concentration of airborne particulate matter in the urban green spaces. The results showed that the following: (1) There were significant correlations between meteorological parameters and particulate concentrations. The concentrations of different sizes of particulate matter in northern China during the heating season are generally higher than those in pre-heating season. In the evening, the concentration of airborne particulate matter was low. At noon, the concentrations of $PM_1$ and $PM_{2.5}$ reached their maximum. The concentrations of $PM_{10}$ and TSP reached their maximum in the morning. It is not recommended to go out during the morning and noon; rather, one should travel less and stay indoors (Figure 9). (2) The effects of different vegetation structures on reducing the concentration of airborne particulate matter with different particle sizes were significantly different, but the effect was consistent at different heights. Partially closed green spaces had strong adsorption capacity for particles with different sizes, which played a certain reduction role. Moreover, the concentration of particulate matter below the canopy was generally higher than that in the middle of the canopy. In order to improve the air quality in the future, the partially closed space enclosure model can be given priority in the planning and design of urban green spaces. For the adsorption of fine particulate matter ($PM_1$ and $PM_{2.5}$), it is recommended to plant the partially closed broad-leaved one-layered forest (PBO). For $PM_{10}$ and TSP, the partially closed broad-leaved multi-layered forest (PBM) and the partially closed coniferous and broad-leaved mixed multi-layered forest (PMM) are recommended. At the same time, the terrain of urban forests can be modified to raise the ground for planting in order to provide potential fresh air during human recreation (Figure 10). The results of this study identified the dust-retention effects of urban green spaces with different spatial structures, which can provide parameterization information for air-improvement-oriented planning and design of urban green spaces in future.

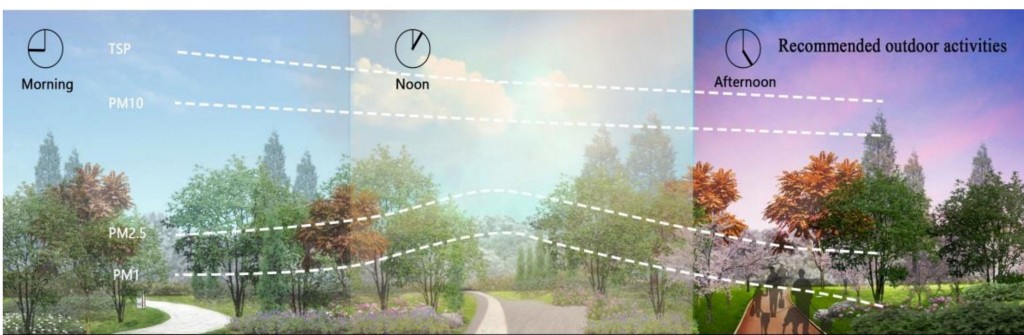

**Figure 9.** Schematic diagrams of time variation of different particle size concentrations.

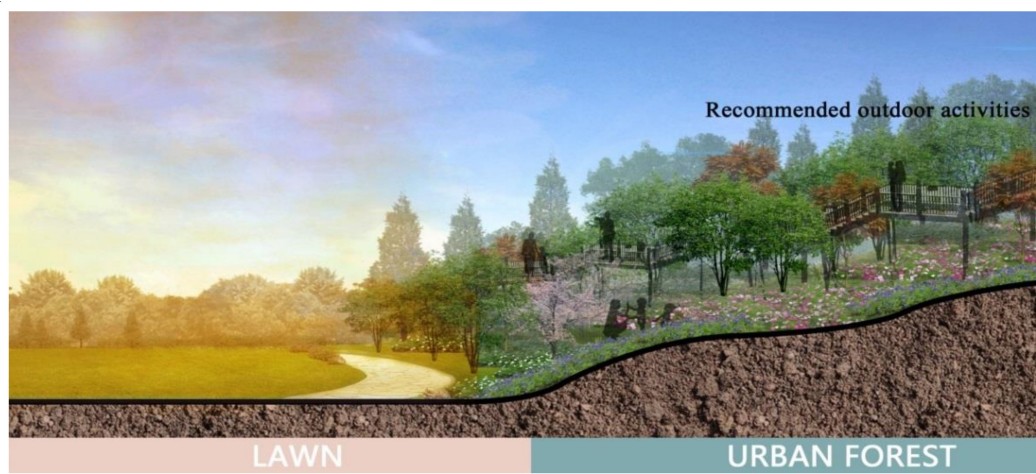

**Figure 10.** Schematic diagram of the influence of vegetation structure on airborne particulate matter concentration.

**Author Contributions:** Conceptualization, L.Q. and T.G.; methodology, L.Q. and T.G.; validation, B.J., L.Q. and T.G.; formal analysis, B.J., L.Q. and T.G.; investigation, B.J., C.S., S.M., Y.C., Z.Z., Y.L., L.Q. and T.G.; resources, L.Q. and T.G.; data curation, B.J., C.S., S.M., Z.Z., Y.C., Y.L., L.Q. and T.G.; writing—original draft preparation, B.J.; writing—review and editing, B.J., L.Q. and T.G.; visualization, B.J., L.Q. and T.G.; supervision, L.Q. and T.G.; project administration, L.Q. and T.G.; funding acquisition, L.Q. and T.G. All authors have read and agreed to the published version of the manuscript.

**Funding:** This research was funded by the National Natural Science Foundation of China (grant number: 31971722), the Science and Technology Innovation Program of Shaanxi Academy of Forestry (grant number: SXLK2021-0216), the Key Program of Science and Technology Innovation of the Shaanxi Academy of Forestry (grant number: SXLK2021-02-0X), the Key Research and Development Program of Xianyang (grant number: 2021ZDYF-SF-0022), and the Scientific Research Cooperation Agreement Project of the Xianyang Forestry Bureau (grant number: 20211221000007).

**Institutional Review Board Statement:** The study was conducted according to the guidelines of the Declaration of Helsinki, and approved by the Institutional Review Board of College of Landscape Architecture and Arts, Northwest A&F University.

**Informed Consent Statement:** Informed consent was obtained from all subjects involved in the study.

**Data Availability Statement:** The data presented in this study are available on request from the corresponding author. The data are not publicly available due to policy of the institute.

**Acknowledgments:** We are grateful to Katie Oswalt of Mississippi State University for helping to revise this manuscript. We also thank the 60 volunteers who helped us with the experiment.

**Conflicts of Interest:** The authors declare no conflict of interest.

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
