# Peer review of "Differences in Airborne Particulate Matter Concentration in Urban Green Spaces with Different Spatial Structures in Xi’an, China"

_forests, doi:10.3390/f13010014_

Round 1

Reviewer 1 Report

The review concerned an article entitled: Difference of Airborne Particulate Matter Concentration in Urban Green Space with Different Spatial Structures in Xi’an, China. Air pollution is still an important topic, especially in places with high pollution where phytoremediation as a solution could be used. Mitigation of air pollution is more and more important this days.  Due to the fact that my native language is not English, I will not check the linguistic correctness of the manuscript. Overall the article is interesting and although it touches on an repeated topic it provides a refreshing approach to research on the phytoremediation.

Just a few questions:

Questions/mistakes/errors,

Line 81 – In my opinion the word dust should be avoided – We are talking the scientific term – Particulate matter

Line 81 – There is lack of information about the new trend in the particulate matter measurement – flowering meadows which are better than lawns. Described in eg. Przybysz et al. 2021. Where trees cannot grow – Particulate matter accumulation by urban meadows.

Line 103 – There is error with the reference.

Line 128 – Do authors wondered about the same types of greenery in other cities for comparison?

Table 1 – The table is not easy to read.

Table 2 – Authors can bold the values which are significant.

Line 205 – Do authors have data about the accumulation of PM on the leaves?

Line 416 – what is the leaf blocking?

Line 433 – This data have 43 years. More research was made in this matter eg. Lukowski et al. 2020 Particulate matter on foliage of Betula pendula… and other

Line 442 – information about flowering meadows is also needed.

Line 446. Some researches says also that mosses can have a big influence in PM accumulation. I thik to the work be complete it should be also added. In eg. Haynes et all. roadside moss turfs in South East Australia capture more particulate matter along an urban gradient than a common native tree species.

Figure 9. I think that this is more for the one city – Xi’an. Its of course important. But other cities ef in Europe in America have different hours of activities. And it will be also depends of the PM sources. I think it should be added in the conclusions.

Reviewer 2 Report

The manuscript entitled “Difference of Airborne Particulate Matter Concentration in Urban Green Space with Different Spatial Structures in Xi’an, China” quantitatively compared the effects of monitoring time, spatial structure and heights of vegetation on the concentration of airborne particulate matter of the urban green spaces. The below comments can improve the manuscript.

- In Table 1, it is better to create a column to add the Latin name of plant species including lawn, grasses, coniferous, broad-leaves and mixed communities.

- There are 5 canopy covers >70% but with different layers and various vegetation types. The results would be more accurate if Leaf Area Index (LAI) for each place or vegetation type was measured. Also, for canopy 40-70%.

- In Lines 471-472: In below sentence,

“In order to improve the air quality in the future, the partly-closed space enclosure model can be given priority in the planning and design of urban green space”. This is below question.

Which vegetation type? PBO? PMB? PMM? Answer can be mentioned in parenthesis after "the partly-closed space enclosure model" in this sentence.
